# Minimum Spanning vs. Principal Trees for Structured Approximations of Multi-Dimensional Datasets

**DOI:** 10.3390/e22111274

**Published:** 2020-11-11

**Authors:** Alexander Chervov, Jonathan Bac, Andrei Zinovyev

**Affiliations:** 1Institut Curie, PSL Research University, F-75005 Paris, France; jonathan.bac@cri-paris.org; 2Institut national de la santé et de la recherche médicale, U900, F-75005 Paris, France; 3CBIO-Centre for Computational Biology, Mines ParisTech, PSL Research University, 75006 Paris, France; 4Centre de Recherches Interdisciplinaires, Université de Paris, F-75000 Paris, France; 5Lobachevsky University, 603000 Nizhny Novgorod, Russia

**Keywords:** data analysis, principal trees, trajectory inference, single-cell transcriptomics, clustering, minimum spanning trees, graph theory

## Abstract

Construction of graph-based approximations for multi-dimensional data point clouds is widely used in a variety of areas. Notable examples of applications of such approximators are cellular trajectory inference in single-cell data analysis, analysis of clinical trajectories from synchronic datasets, and skeletonization of images. Several methods have been proposed to construct such approximating graphs, with some based on computation of minimum spanning trees and some based on principal graphs generalizing principal curves. In this article we propose a methodology to compare and benchmark these two graph-based data approximation approaches, as well as to define their hyperparameters. The main idea is to avoid comparing graphs directly, but at first to induce clustering of the data point cloud from the graph approximation and, secondly, to use well-established methods to compare and score the data cloud partitioning induced by the graphs. In particular, mutual information-based approaches prove to be useful in this context. The induced clustering is based on decomposing a graph into non-branching segments, and then clustering the data point cloud by the nearest segment. Such a method allows efficient comparison of graph-based data approximations of arbitrary topology and complexity. The method is implemented in Python using the standard scikit-learn library which provides high speed and efficiency. As a demonstration of the methodology we analyse and compare graph-based data approximation methods using synthetic as well as real-life single cell datasets.

## 1. Introduction

Graph theory based methods play a significant role in modern data science and its applications to various fields of science, including bioinformatics. One of the applications of graphs in unsupervised machine learning is for dimensionality reduction, where the geometrical structure of a data point cloud is approximated by a system of nodes embedded in the data space and connected by edges in a more or less complex graph. For example, this graph can represent a regular grid as in Self-Organizing Maps (SOM) [1], Elastic Maps (ElMap) [2,3] or principal curve approaches [4,5]. In this case, the graph can serve as a base for reconstructing a low-dimensional principal manifold [3,6]. However, in some applications of graph-based approximation, the geometry of a dataset can be characterized by more complex structures such as branching points, excluded regions and regions of varying local intrinsic dimensionality [7,8,9]. For example, some large-scale clinical or single cell omics datasets can be modeled as a set of diverging and bifurcating trajectories connecting a root state (e.g., a progenitor cell) with a number of final states (e.g., corresponding to fully differentiated cells). In this case, a more complex graph than regular grid graphs should be used for efficient data approximation. Graph-based data approximation (GBDA) constructs a “data skeleton” which can be used to simplify, compress and denoise the complete data description.

A challenging question in GBDA is to define the appropriate graph topology, which should be more complex than a regular grid and less complex than a general graph with arbitrary topology (in the latter case we might end up with a graph as complex as the data themselves). One evident example of such a compromise consists in approximating the data point cloud by a graph with tree-like topology. A more advanced approach consists in introducing cubic complexes as more sophisticated graph types [10]. The task of approximating a data point cloud by a graph can be split into two interconnected subtasks. The first subtask is to define optimal (in some mathematically defined sense) embedment of graph nodes in the multi-dimensional space, when the graph topology is fixed. The second, much more challenging subtask is to find the optimal (in some sense) graph topology matching the dataset structure. Several mainstream ideas have been suggested for solving these problems: one consists in exploiting the properties of Minimal Spanning Tree (MST) [11] and another consists in building principal graphs (PG) [12]. The latter approach can be either equipped with a systematic search for the optimal graph topology as in the topological graph grammar approach [10] or using some heuristics to make an intelligent guess on the appropriate topology as in SimplePPT [13] or so called reversed graph embedding approaches [14].

One particular important application of GBDA approach is the computational data analysis of single-cell omics datasets [15]. A number of new methods have been developed in this field combined under the name cellular trajectory inference (TI) [16]. TI methodology provides a useful tool for studying cell differentiation, cell cycle and other central biological processes with applications in cancer [17] and developmental biology [18]. These methods essentially create a graph-based approximation of a data point cloud where each data point represents a molecular profile of a single cell allowing one to arrange the cells along the graph edges and branches, locally ranking the cells and thus obtaining insights into the underlying biological processes. The availability and the complexity of single cell omics datasets serves a source of motivation for us here.

The present study is devoted to improving and benchmarking existing GBDA approaches. The question of benchmarking the GBDA methods is significant since, for example, only in bioinformatics dozens of new GBDA methods have been introduced in the recent years. Most of them build linear, tree-like or simple circular graph topologies but among these methods there exist those building more general graph types (such as partition-based graph abstraction (PAGA) [19]). It is desirable to understand their advantages and disadvantages and to formulate rules for choosing the most appropriate ones for a given dataset or datatype (e.g., gene expression vs. other types of experimental assays). This question has been mentioned in a recent review “Eleven grand challenges in single-cell data science” [15]: “A pressing challenge is to assess how the different trajectory inference methods perform on different data types and importantly to define metrics that are suitable.” Recently several studies appeared devoted to systematic comparison of trajectory inference methods in bioinformatics of single cells [16].

We contribute here to the question of comparing various graph-based data approximators following an approach which was not previously exploited. The approach presented here can not only be used to compare several GBDA methods, but also to fine-tune the hyper-parameters of any GBDA method, thus moving forward the GBDA methodology. In previous benchmarking attempts, the topological structures of graph-based approximators have been compared directly [16,20,21]. Quite opposite, the principle of our approach is reducing the problem of comparing two approximators to the well-established realm of comparing clustering methods. The idea of this reduction is the following: As a first step, subdivide an approximating graph into segments; as a second step, cluster the data point cloud using these segments, i.e., assign each data point to the nearest graph segment (Figure 1). Each approximating graph topology and geometry “induces” certain clustering (data point partitioning) of a dataset. At the third step we suggest using the well-established methods to compare two clustering results (such as Rand or other scores). Since the induced clusterings are based on the graphs, one thus obtains a score how different/similar these graph approximations are. In addition, at the last step one can use the standard clustering quality scores (such as silhouette [22]) which measures how good is the clustering itself. In this way, one can adjust the (meta)-parameters of the graph approximation methods choosing those with the largest clustering quality score.

The graph segments mentioned above may be thought as connected components of a graph without any branching point. The exact definition is provided below in the main text. The proposed scores possess the main desired property for GBDA method benchmarking: They depend on “trajectories” rather than on “graphs”, meaning that adding new nodes or rearranging them will not change the scores if “trajectories” remain unmodified.

We argue that the proposed methodology provides a convenient scoring for comparing GBDA approaches as well as for parameter fine-tuning. It is implemented using Python sklearn library which provides highly standardised and efficient set of tools for calculation clustering quality and clustering comparison scores. In order to demonstrate the effectiveness of our approach we compare two mainstream GBDA methods, one is based on minimal spanning trees and another one based on application of elastic principal graphs.

## 2. Results

### 2.1. Comparing and Benchmarking Graph-Based Data Approximation Methods Using Data Point Partitioning by Graph Segments

In order to benchmark the graph-based data approximation (GBDA) methods, a standard strategy consists in creating synthetic datasets with known underlying ground truth graph topology as a generative model, applying the methods and comparing how far the reconstructions are from the ground truth graph model.

The contribution of the current paper is an alternative approach for comparison of two graph-based approximations. Our proposal has certain advantages comparing with the approaches used before. First of all it compares “trajectories” rather than “graphs” meaning that (omitting the technical details) adding more nodes to the graph (or rearranging them), but preserving the same “trajectories” does not change scores. It is easy to implement and interpret it for graphs of arbitrary complexity, while many other approaches has limitations or unclear interpretations on complex graph structures. Roughly speaking the main idea is to reduce the complex graph comparison problem to the well studied problem of clustering results comparison.

#### 2.1.1. Outlining the Method

In order to compare (i.e., calculate similarity score) two graphs approximating given dataset, we perform three steps:Split each approximating graph into segments. By a segment we mean certain path from one branching point or leaf node to another.Create clustering (partitioning) of the dataset by segments. Each data point is associated with the nearest segment of the graph. Thus, in general, if graph is partitioned into *N* segments at the previous step then the dataset will be partitioned into *N* clusters.Compare the clusterings for the dataset using standard metrics (like adjusted Rand index or adjusted mutual information or any other suitable measure). Since the clusterings are produced from the structure of the approximating graphs, the score shows how two GBDA results are similar to each other.

Figure 1 illustrates steps 1 and 2. More detailed description of all three steps is provided below. The Python code and Jupyter notebooks are described in the “Implementation” section.

#### 2.1.2. Details on Step 1: Segmenting the Graphs

Roughly speaking we split the graph into maximal non-branching segments. We can obtain such splitting by throwing out all branching points and taking the connected components of the obtained graph as desired segments, modulo rejoining back dropped branching points and some other details. Recall that branching node is node of degree higher than two, and leaf (or terminal node) is node of degree 1. Formally speaking let us call a segment of a graph a path in the graph which starts at a branching or a leaf node and ends also at a branching or a leaf node, that does not contain any other branching point except those possible at the ends. As one can see, such definition reflects the intuition underlying the notion of the “segment”: however, we need to specify several exceptional cases. For a graph which is an isolated cycle (not containing branching or leaf nodes), the whole cycle will be considered as a “segment”. The same is true if a graph contains several connected components which are cycles: Then all of them are considered as separate “segments”. The other exceptional case is case of nodes of degree zero (isolated nodes); they also will be considered as separate “segments”. These exceptional cases cannot happen for connected principal trees which are the main object of the present study, they are just mentioned here for completeness.

It is not difficult to prove that: Any graph can be uniquely partitioned into such type of segments.

One can get the proof by a mild formalization of the argument with connected components mentioned above. The other way is to describe a constructive process to obtain such partitioning of the graph that is done in our Python implementation of the method.

We coded in Python a version of the depth-first search algorithm to produce a split into segments for an arbitrary graph. The main difference to the classical depth-first search is a storage of visited edges (not only nodes) of the graph to correctly process possible cycles in the graph. The algorithm starts from any branching or leaf node, and walks along edges in depth, joining them to the “current segment” until it meets a branching or a leaf node. Here, the “current segment” is terminated. In case of a leaf node one returns from the recursion, the same for already visited branching node. In case of a new branching node (not visited before) one goes into deeper level of depth-first recursive process.

#### 2.1.3. Details on Step 2: Clustering (Partitioning) Data Points by Graph Segments

Having obtained an approximating graph split into several segments (see the previous step), we can cluster the dataset by these segments. This means that for each datapoint we find the nearest segment. There is slight ambiguity how that can be done, however that ambiguity would not much effect the results (at least for the graphs quite many nodes) and so, for our purposes we choose the simplest way to define the proximity. We calculate the distances between all datapoints and all nodes of the graph. If the nearest node belongs to only one segment then the datapoint gets it as a label. If there are several segments to which the nearest node belongs, then we take the second nearest node (among nodes from the selected segments) and label the data point by the segment to which that second nearest node belongs to. The algorithm is implemented in Python, Figure 2 illustrates its work. The alternative way to define the proximity would be for each datapoint calculate distance not only to nodes, but also to find nearest point on the graph (which might be somewhere at some edge, not node) and since all edges belong to segments—that segment give a label to that datapint. That way is theoretically better, however practically computationally more constly and the difference between the two methods would not be big for datasets of our primary concern—single cell data—which are quite noisy. Thus in what follows we consider the proximity defined as above. Let us also mention that with any way to define the proximity there can be some ambiguity—some points may be on equal distance from several segments; however, the set of such points would typically be of some measure zero, i.e., practically such cases would be extremely rare and we can make arbitrary choice in such situations.

#### 2.1.4. Details on Step 3: Compare the Clustering Measures for the Datasets

Having produced clustering of the dataset by several approximating graphs (see previous steps) one can use the well-established and efficiently implemented in Python sklearn library scores for clustering comparison. From our construction it follows that if the graphs are similar to each other then the clustering will be similar also, and the scores will reflect it, while dissimilar graphs will lead to dissimilar clusterings and the scores will show it. Thus these scores give efficient measure of similarity for graphs, for both geometry and topology at the same time.

There are many scores used to compare clustering: Adjusted Rand index [23], adjusted mutual information [24], Fowlkes-Mallows score [25], V-measure [26], etc. According to our experiments in most cases all the scores show similar results. So as a default choice we would suggest adjusted Rand index. Let us mention that “adjusted” (adjusted Rand index, adjusted mutual information) indicates a modification of the score such that for two random data clustering the score becomes near zero, that is a desirable property and such score modifications should be used by default.

Figure 2 shows examples of graph comparison. One can see that all the scores lead to the same conclusions which perfectly fit the intuition.

The suggested approach requires a dataset for which the compared graphs serve as an approximation, thus it does not allow to compare graphs on their own. It might seem as a disadvantage, but actually it is a way around for main applications, since the approach would give a more focus and weight for the regions of the data space where more data points are concentrated.

### 2.2. Unsupervised Scores for Comparing GBDA Methods and Parameter Fine-Tuning

Another application of our approach is to provide a score for how good a given approximating graph is for a given dataset. In real situation there is no “ground truth” so having several GBDA methods (and typically many hyper-parameters for each of them) it is not always easy to choose which is the best one. Our suggestion provides certain insights how the question can be approached. The main idea is to transfer the methods used to score clustering to the GBDA area. We do not claim that the method always works very well, but we will demonstrate that it makes sense at least in some cases. That situation is similar to clustering setup, where choice of appropriate cluster number or a clustering method is frequently a subtle question for real-life datasets.

The approach is very similar to what is described in the previous section. To calculate a score of goodness of a given approximating graph for a given dataset we perform three steps:Split each approximating graph into segmentsCreate clustering (partitioning) of the dataset by the segmentsUse standard metrics and tools to calculate how good is the clustering:, e.g., calculate silhouette, Calinski–Harabasz, Davies-Bouldin score etc.

The first two steps are exactly the same as described in details in the previous section, so we do not comment on them here. The last step is standard for clustering techniques: There are several scores which provide certain estimates how good a clustering is. They are efficiently implemented in Python sklearn library. It should be noted that all three indices work better for convex clusters, rather than for other cluster shapes—that is a disadvantage in certain situations. Silhouette score is bounded from −1 to 1, with higher values indicating better clustering, Davies-Bouldin index is non-negative, with zero indicating perfect clustering, Calinski–Harabasz index is positive and non-normalized with higher values indicating better clustering. Figure 3 shows an example: for a given dataset we consider three approximating graphs. It is intuitively quite clear from the figure that the first produces the best GBDA of the dataset, while the last one is the worst. Silhouette and Davies-Bouldin scores fit the intuition. Calinski–Harabasz does not work perfectly even in this simple example, which illustrates that the approach should be used with some care. We also provide a plot–silhouette score dependence on the node numbers of approximating graphs, it also confirms the conclusions above—lower score is assigned for graphs with higher nodes (which are constructed by MST technique)—which is natural since these graphs have much more branching points than underlying ground truth dataset.

In what follows we consider the proposed method in more details: in Section 2.5 we work out in details use case for the proposed method; in the section devoted to single-cell example we apply that strategy to real data—as one can see it allows quite successfully to determine the range of the reasonable parameters; the plot in the next section presents results of simulations on thousands of artificial datasets, we can see that maximum of silhouette is unfortunately not exactly at the point of optimal parameters but nevertheless it is quite near.

To conclude, we proposed a method to choose better parameters for trajectory constructions—to choose those parameters where proposed scores are reasonably high. Our analysis suggests that there is no always guarantee that exactly the maximum of scores corresponds to the best parameters, however in many situations best parameter are not far from that point. So for practical purposes our method allows to restrict the range of parameters and that restricted range might be analysed additionally, using for example domain knowledge.

### 2.3. Comparison of Clustering Based Scores with Other GBDA Metrics

In this section we show that the proposed scores can be matched to some simple and natural GBDA metrics (such as the number and the position of branching points). Such simple scores are good only for comparison of graphs in quite simple situations and their generalization to more complex cases is non-evident, if possible. Compatibility of the clustering-based scores with the natural scores provides one more justification that our approach quite corresponds to expected intuition for trajectory comparison.

In order to make such a comparison, we created synthetic datasets with clear and simple ground truth graph structure, representing a binary tree (see Figure 4A).

Since the ground truth graph is quite simple and it can be characterized by a number of branching and their positions, so we can calculate these values and compare them to the ground truth. It provides us the natural graph metrics. Of course, one cannot compare the number of branching points (which is integer) with the adjusted Rand index (which is continuous number between 0 and 1). What one can compare is if both scores suggest the same type of graph in the family to be the best approximation of the ground truth.

We run a simple GBDA algorithm based on MST applied to such a dataset. The main parameter of such algorithms is the number of nodes in the graph. After that we looked what is the optimal value of this parameter suggested by different scores:, i.e., the scores based on adjusted Rand index, mutual information and naive scores based number of branching points and their positions. The main observation is that results perfectly correspond to each other (see Figure 4B). Each point on these plots represents an averaged value of scores over one hundred random datasets. Our experiments showed that performing one hundred random simulations is more than enough to get statistically significant conclusions.

From these examples one can see difficulties using simple scores to compare approximating graphs: if the ground truth graph has three branching points, and the approximating graph has different number of branching points—how one would compare them? We address this question in our simulation in the following way: Take 3 nearest branching points of the approximating graph to the branching points of the ground truth graph and neglect other possibly present branching points. If there is less than 3 branching points in the approximation then one branching point serves as the nearest to several in the ground truth graph. We can do this because we see that for our best parameter value, number of nodes = 10, the number of branching points equals to 3, which is the same as for the ground truth graph. Thus the question above is avoided for that particular example, but it cannot be avoided for more complicated datasets.

### 2.4. Comparison of MST-Based Graph-Based Approximations and ElPiGraph

In that section we apply the technique introduced above to compare two popular GBDA techniques. The first one is a simple method based on application of MST (minimal spanning trees) and the second one is the ElPiGraph (Elastic Principal graphs). The simple idea of computing MST underlies many popular methods for cellular trajectory inference [16,27,28,29], this is why it is a natural choice to compare with ElPiGraph representing principal graph approach combined with topological graph grammar-based topology optimization. A brief description of both methods is provided in “Materials and Methods” section.

The main conclusions and highlights of the section are the following:Both MST and ElPiGraph can achieve similar quality of graph-based data approximation by tuning their main parameter, number of nodes. The quality here is estimated by the scores introduced above and in comparison with ground truth for simulated datasets. However, the advantage of ElPiGraph is in its stability with respect to this parameter. Small modifications do not change the quality dramatically, while for MST even small modifications of the parameter may lead to quite a drastic change of the approximating graph structure and, respectively, the quality of ground truth approximation. That means that in practical situation where ground truth is not available, ElPiGraph has an advantage since one should not be afraid of choosing the parameter incorrectly. Therefore, a reasonably wide range of parameters gives similar results, while for MST approach choosing an incorrect parameter would give a significant loss in approximation quality.The embedment of approximating graphs produced by ElPiGraph are much smoother comparing to MST. This observation is consistent with the previous one and with ElPiGraph algorithm which penalizes for non-smoothness and hence it produces graphs which would not produce too much branching points, the problem which underlies the MST instability.We analyse a possible combination of MST+ElPiGraph methods, where the initial graph approximation is obtained by MST and then ElPiGraph algorithm starts from this initialization. One of our conclusions is that ElPiGraph “forgets” the initial conditions quite fast. Nevertheless the combination of methods may have certain advantages.We consider a real-life single cell RNA sequencing dataset and demonstrate that our approach for choosing the parameters of the trajectory inference method based on unsupervised clustering scores provides reasonable results.

#### 2.4.1. Smoothness of ElpiGraph-Based Graph Approximators Comparing to MST

One of the features of ElPiGraph is the smoothness of its trajectories.

Figure 5 shows examples of ElPiGraph trajectories (green color). (See more pictures in the section “Materials and Methods”). Even in these simple examples it is evident that ElPiGraph provides much smoother trajectories which give the correct branching structure for much wide range of parameters comparing to MST (black colored graphs).

It is not surprising since its trajectories are defined by minimizing the functional containing the penalty for non-smoothness. This feature is related to stability of ElPiGraph with respect to parameters, especially compared to MST. Figure 5 shows examples of MST and ElPiGraph trajectories for the same node numbers. One can evidently see ElPiGraph trajectories are much smoother, and able to reproduce correct branching structure for much wider range of node numbers. Smoothness and stability are quite related, since non-smoothness of MST trajectories leads to creation of small incorrect branches of the graph, which does not happen for ElPiGraph.

#### 2.4.2. Comparison between ElPiGraph and MST on Large Number of Different Datasets

Consider several types of datasets with clear ground truth graphs, for each type of dataset create one hundred sample datasets differing by random fluctuations in the generative model. The approximating graphs are produced for each dataset by both MST and ElPiGraph methods and compared via the proposed methodology to the ground truth. Then the scores are averaged within each type of dataset to get statically reliable result. Moreover both MST and ElPiGraph are simulated for many values of the main input parameter, number of nodes. Thus we get the plots showing scores of each method depending on the number of nodes.

The conclusions are the following, for most of the dataset types both methods can achieve approximately the same highest score. However, the main advantage of ElPiGraph is much higher stability. Its dependence on the input parameter is much less pronounced, which in practice (where no ground truth is known) is a crucial advantage, since a rough estimate of the parameter is enough to get almost top quality. For the MST-based approach a small error in the estimation of the parameter may lead to quite a drastic change of the approximation quality. To the best of our knowledge there is no universal method to estimate the required node number in the approximating graph. Sometimes one uses the following heuristic: The node number is approximately square root of the number of points in the dataset. Our analysis below shows that this choice is not always close to optimal, i.e., the number of nodes corresponding to the highest score can be quite far from the suggestion of this heuristic.

Figure 6 demonstrates the results of the simulations. One can see that blue plots (ElPiGraph) are much flatter than red ones (MST) which shows higher stability of ElPiGraph. We do not show all types of the datasets and all scores, in order not to overload the figures. Only plots for four types of datasets, for adjusted Rand score and for number of branching points are presented. The results for other dataset types and other scores are quite similar and does not change the conclusions.

#### 2.4.3. Initialization of ElPiGraph with MST-Based Tree

In the default scenario, the ElPiGraph algorithm builds the approximating graph in an iterative manner, adding nodes one by one. Thus it is possible to initialize it with a graph obtained by some other algorithm. For example, ElPiGraph can be initialized with a tree built using an MST-based approach. Indeed, such an initialization is implemented in a popular cellular trajectory inference tool STREAM [20], and other tools use different heuristics for initializing ElPiGraph [30].

Here we consider the combination of MST and ElPiGraph algorithms within one method. One creates the MST graph with *K* nodes and initializes ElPiGraph algorithm with this input, and constructs the elastic principal tree with K+N nodes. From the conducted computationals experiments, the main conclusion was that ElPiGraph forgets the initialization rather rapidly. That is yet another argument for stability of ElPiGraph: in examples below we take as initial condition MST graph with “overbranching” (i.e., much more branching points than in the ground truth graph), and ElPiGraph algorithm in several steps creates a graph with correct number of branching points.

In order to demonstrate our conclusions, we run simulations over various test datasets. We compared constructions by MST, ElPiGraph and several combinations of MST and ElPiGraph differing by the number of nodes in the initializing MST graph. We compared different scores developed in the previous sections for constructed graphs. And we see that statistically averaged scores over many simulations confirm our observation that ElPiGraph quite rapidly forgets the initialization.

Figure 7A,B give one example of such behavior, ElPiGraph initialized by 30 node MST graph in 5 steps produces a graph similar to the case when ElPiGraph was applied without initialization by MST, despite that initialization was drastically different: it had a large number of branching points and was quite non-smooth. Figure 7C confirms similar phenomena in statistical manner: each point of each plot is result of averaging over hundred of simulated tree-like datasets. Result for other dataset types and other scores are similar. Combination of MST and ElPiGraph can be useful in some cases, for example for large number of nodes current implementation of MST works faster, so one can first create a “draft” graph by MST and then “polish” it by ElPiGraph, thus gaining faster implementation.

### 2.5. Tuning Parameters of GBDA Algorithms Using Unsupervised Clustering Quality Scores

Let us demonstrate that the proposed approach to choose best parameters of a GBDA algorithm based on unsupervised clustering scores (silhouette, Calinski–Harabasz, Davies-Bouldin) provides reasonable results in certain situations. Once again, let us consider a simulated dataset with clear ground truth graph structure (e.g., a binary tree with 3 branching points), random normal noise with standard deviation 5 is added, the dataset is built in 20 dimensional space, the direction of each edge is chosen randomly.

The MST-based GBDA algorithm is applied, and the constructed tree is compared with ground truth graph using the adjusted Rand index-based score and also scored by unsupervised score (not requiring the ground truth). MST construction depends on the node number parameter. One can see (Figure 8) that for all unsupervised scores the range 20-40 of the parameter gives almost highest scores, thus that range is quite a good choice of the parameter. It is quite in agreement with supervised score based on adjusted Rand index - which achieves it maximum for 40 nodes. And also we can check that number of branching points is 3 at this range as for the ground truth. Thus we see unsupervised scores provide reasonable suggestion for the choice of the parameter. The only thing we should mention is that we see some incorrect maximum for the unsupervised scores for very small value of parameter, e.g., 6 for silhouette, however that maximum is not stable. So to conclude one should use unsupervised scores with certain care, not just taking the maximum value, but checking its stability and if possible some subject area (e.g., biological) meaning.

### 2.6. Single Cell Data Analysis Example

Cellular trajectory inference for single cell RNA sequencing data is important research direction in modern bioinformatics [15,16]. Since it can provide biological insights into important processes: cell differentiation, cell cycle and development of pathology like cancer. In the present section we demonstrate our approach on a single cell transcriptomic dataset from [31]. The data describes 24,748 gene expressions for 447 cells from liver of mouse embryos. The process under consideration is differentiation of bi-potential hepatoblasts into hepatocytes and cholangiocytes (two main different cell types in the liver). Figure 9A shows PCA projection and indicates regions for these three cell types.

The main goal here is to confirm conclusions made in the previous sections from the analysis of synthetic data by consideration of a real single-cell example.

Namely:ElPiGraph demonstrates much higher stability than MST and choice of main parameter (node number) can be made more easily and reliably.Unsupervised metrics for trajectories introduced above (based on silhouette, Calinski–Harabas, Davies-Bouldin clustering scores) provides insights for the choice of parameters for trajectory inference methods.Heuristic for MST, to take number of nodes equal to approximately square root of number of datapoints fails significantly for the present example. It would give 21 nodes, however trajectories of MST with 21 nodes show significant “overbranching” (see figures below). Reasonable node numbers for MST is 9 or a little below, but 10 and above leads to biologically incorrect branching.

In order to make these conclusions we constructed approximating graphs using MST and ElPiGraph approaches for different node numbers and calculated the unsupervised metrics proposed above. Both methods were applied after PCA-based linear dimensionality reduction to 50 dimensions since such use of PCA is standard preprocessing for single cell datasets. Biological background suggests that the correct trajectory should have exactly one branching point, since the process is differentiation of bi-potential hepatoblasts into two cell types (hepatocytes and cholangiocytes). Thus trajectories with large number of branching points are biologically incorrect. Figure 9B,C show that ElPiGraph method produces reasonable trajectories for 5–22 nodes, while MST only for 5–9 nodes, so ElPiGraph is much more stable. Using two unsupervised metrics (silhouette and Davies-Bouldin) we can very clearly distinguish the threshold node number 23, where the overbranching for ElPiGraph first appears (Figure 9B). Indeed these metrics abruply jumps at this point—see Figure 10B. Thus confirming our approach to use unsupervised metrics to tune parameters. (From Calinski–Harabas score it is less evident in that example). Using unsupervised scores for MST trajectories one can also get an indication for the correct values of the node number parameter, the threshold value is 10 nodes, where biologically incorrect branch first appears. While that indication is not fully perfect nevertheless one can see significant drop of scores beyond 9 nodes (Figure 10A). Moreover at 9 nodes there is local extremum for all scores, which is standard indication for parameters choice.

## 3. Discussion

In the paper we proposed an improvement for benchmarking and parameter tuning for graph-based data approximation methods. The principle of the approach is in transferring the methods developed for comparing the clustering results to the trajectory inference area. We performed multiple simulations and tests to argue that the approach is efficient. In particular we made a detailed comparison of GBDA methods based on MST (minimal spanning tree) and ElPiGraph (Elastic Principal Graph). We conclude that ElPiGraph is much more stable and parameter tuning can be made more easily and reliably for it in the absence of the ground truth knowledge.

Let us remark that the methods of graph-based data approximation are applicable in the data spaces of arbitrary ambient dimensionality. The examples used in this article to define and illustrate the suggested approach are two-dimensional but the methods can be applied also in higher dimensions without any modifications, as demonstrated in the analysis of a real-life single cell dataset. Nevertheless, extracting trajectories from data point clouds assumes that the underlying point density function is characterized by the existence of continuous one-dimensional “ridges” which can diverge from or converge to each other in the data space. They can also connect local density peaks. In this case, the appropriate data approximation methods (such as ElPiGraph) look for principal curves and, more generally, branching principal trajectories, along which the data points are condensed. In the case when the data point clouds are not characterized by such a structure, when the local intrinsic dimension is much larger than one, extracting trajectories using principal graphs or any other types of methods is meaningless and leads to arbitrarily defined graphs, which are pointless to compare. Evaluating the general applicability of the graph-based data approximation methodology will make a subject of our future work.

The methodology suggested in this study should find applications in such fields as trajectory-based analysis of single cell datasets, defining dynamical phenotypes in large scale observational data (e.g., clinical data), learning the patterns of motion of complex objects (such as human), skeletonization of images and other fields where large multi-dimensional data point clouds are characterized by a complex intrinsic organization.

## 4. Materials and Methods

### 4.1. Minimal Spanning Tree-Based Data Approximation Method

Minimal Spanning Tree (MST) for a graph is a tree-like subgraph which contains all nodes of the original graph. For a weighted graph one requires minimality of the tree in the sense that sum of weights on the edges of a tree is minimal. MST is one of the central concepts and tools in graph theory [11]. It is widely used in various trajectory inference methods such as Waterfall [27], TSCAN [28], Slingshot [29]. Let us remind the main ideas, provide simple implementation and examples.

Essential steps for many approaches are the following steps:Split the dataset to certain pieces (for example, by applying *K*-means clustering for some *K*).Construct kNN (*k*-nearest neighbour) graph with nodes at the centers of clusters.Compute an MST (minimal spanning tree) of the kNN graph computed as described above.

There are various modifications around these ideas used by different authors. For our purposes we will consider just the simplest algorithm which directly implements the steps above. The main parameter of the algorithm, number of nodes in the graph, which is the same as number of clusters for K-means clustering. Another parameter is the choice of “k” for the kNN graph, but results does not essentially depend on a large enough “k”, so if computation speed is not an issue one may take the maximal possible “k” equal to the number of input points, i.e., just to consider the fully connected weighted graph. We provide notebooks with implementations of this algorithm in Python.

Figure 11A illustrates the work of the algorithm. One can see that it works reasonably well, however, not so perfect even on examples which might be simple even for a visual-based analysis. There is evident role of the principal parameter, the number of nodes in the graph. If it is too small, then it is impossible to catch the branching structure of the dataset. If it is too large then the algorithm “overbranches”, i.e., it creates branches which do not correspond to the ground truth. Correct choice of the node number parameter seems to be a subtle point for MST-based method.

### 4.2. Method of Elastic Principal Graphs (ElPiGraph)

Elastic principal graphs are graph-based data approximators [3,6,10,12]. We provide its brief description here for self-contained reading. A detailed description of ElPiGraph and related elastic principal graph approaches is available elsewhere [32].

The graph nodes are embedded into the space of the data, minimizing the mean squared distance (MSD) to the data points, similarly to the *k*-means clustering algorithm. However, unlike unstructured *k*-means, the edges connecting the nodes are used to define the elastic energy term. This term is used to create penalties for edge stretching and bending of segments. To find the optimal graph structure, ElPiGraph uses topological grammar (or, graph grammar) approach and gradient descent-based optimization of the graph topology, in the set of graph topologies which can be generated by a limited number of graph grammar operations.

Formally, elastic principal graph is an undirected graph with a set of nodes $V=\{V_{i}\}$ and a set of edges $E=\{E^{i}\}$. The set of nodes *V* is embedded in the multidimensional space. In order to denote the position of the node in the data space, we will use the notation $\phi (V_{j})$, where $\phi (V_{j})$ is a map $\phi :V\to R^{m}$. The optimization algorithm search for such ϕ() that the sum of the data approximation term and the graph elastic energy is minimized. The optimization functional is defined as:(1)$$U^{\phi }\left(X,G\right)=MSD^{\phi }\left(X,V\right)+U_{E}^{\phi }\left(G\right)+U_{R}^{\phi }\left(G\right),$$
where
(2)$$MSD^{\phi }\left(X,V\right)=\frac{1}{\left|X\right|}\sum_{i=1}^{\left|X\right|}\min(||X_{i}-\phi \left(V_{P(i)}\right){\left|\right|}^{2},R_{0}^{2}),$$
(3)$$U_{E}^{\phi }\left(G\right)=\sum_{E^{i}}\lambda_{penalized}\left(E^{i}\right){\left(\phi \left(E^{i}\left(0\right)\right)-\phi \left(E^{i}\left(1\right)\right)\right)}^{2},$$
(4)$$U_{R}^{\phi }\left(G\right)=\mu \sum_{S^{j}}{\left(\phi \left(S^{j}\left(0\right)\right)-\frac{1}{deg(S^{j}\left(0\right))}\sum_{i=1}^{deg(S^{j}\left(0\right))}\phi \left(S^{j}\left(i\right)\right)\right)}^{2},$$
(5)$$\lambda_{penalized}\left(E^{i}\right)=\lambda +\alpha \left(\max(2,deg\left(E^{i}\left(0\right)\right),deg\left(E^{i}\left(1\right)\right))-2\right),$$
where |V| is the number of elements in set *V*, $X=\{X_{i}\},i=1,\cdots ,|X|$ is the set of data points, $E^{i}\left(0\right)$ and $E^{i}\left(1\right)$ denote the two nodes of a graph edge $E^{i}$, star $S^{j}$ is a subgraph with central node $S^{j}\left(0\right)$ and several (more than 1) connected nodes (leaves), $S^{j}\left(0\right),\cdots ,S^{j}\left(k\right)$ denote the nodes of a star $S^{j}$ in the graph (where $S^{j}\left(0\right)$ is the central node, to which all other nodes are connected), $deg(V_{i})$ is a function returning the order *k* of the star with the central node $V_{i}$, and $P\left(i\right)={\mathrm{argmin}}_{j=1,\cdots ,|V|}{∥X_{i}-\phi \left(V_{j}\right)∥}^{2}$ is a data point partitioning function associating each data point $X_{i}$ to the closest graph node $V_{P(i)}$. $R_{0}$, λ, μ, and α are parameters having the following meaning: $R_{0}$ is the trimming radius such that points further than $R_{0}$ from any node do not contribute to the optimization of the graph, λ is the edge stretching elasticity modulo regularizing the total length of the graph edges and making their distribution close to equidistant in the multidimensional space, μ is the star bending elasticity modulo controlling the deviation of the graph stars from harmonic configurations (for any star $S^{j}$, if the embedding of its central node coincides with the mean of its leaves embedding, the configuration is considered harmonic). α is a coefficient of penalty for the topological complexity of the resulting graph.

Given a set of data points and a principal graph with nodes embedded into the original data space, a local minimum of Uϕ(X,G) can found by applying a splitting-type algorithm. At each iteration given the initial guess of ϕ, the partitioning P(i) is computed, and then, given the P(i), Uϕ(X,G) is minimized by finding new node positions in the data space. The convergence of this algorithm is proven [12,33].

Topological grammar rules define a set of possible transformations of the current graph topology. Each transformation produces a new graph topology which is fit to the data. The topology possessing the minimal energy Uϕ(X,G) after fitting the candidate graph structures to the data is chosen as the locally best with a given number of nodes. Topological grammars are iteratively applied to the selected graph until given conditions are met (e.g., a fixed number of grammar application, or a given number of nodes is reached, or the required approximation accuracy $MSD^{\phi }\left(X,V\right)$ is achieved). The graph learning process is reminiscent to a gradient descent-based optimization in the space of all possible graph structures achievable by applying a set of topological grammar rules (e.g., in the set of all possible trees).

One of the simplest graph grammars consists of two operations ’add a node to node’ and ’bisect an edge’, which generates a discrete space of tree-like graphs [32]. The resulting elastic principal graphs are called elastic principal trees in this case.

Any vector in the dataspace *x* – not necessary belonging to the dataset *X* – can be projected onto the tree and mapped onto one of the nodes or an edge in between two nodes. The projection is achieved by finding the closest point on the principal graph as a piecewise linear manifold, composed of nodes and edges as linear segments connecting nodes.

The ElPiGraph package implemented in Python is available from https://github.com/sysbio-curie/ElPiGraph.P. ElPigraph serves as the algorithmic core for several methods of cellular trajectory inference [20,30] and for quantifying trajectories from synchronic clinical data [34].

In the examples used in this article, the principal tree inference with ElPiGraph was performed using the following parameters: $R_{0}=\infty$, α=0.01, μ=0.1, λ=0.01.

### 4.3. Benchmark Implementation

The paper is supplemented with Python notebooks which provide relevant code see https://github.com/chervov/MSTvsET. For better reader convenience we also stored notebooks and single-cell data at Kaggle environment https://www.kaggle.com/alexandervc/trajectory-inference-single-cell-rna-seq/notebooks. On Kaggle one can not only store code, but also run code, store datasets, version control and various social networking possibilities, comment and discuss code, easy forking notebooks etc. Notebook “GraphSegmentation and DataClusteringByGraphSegment” provides main functions from our package “find branches”, splitting the graph into segments and “branch labler”—clustering dataset by graph segments. Used examples are given there. Notebook mostly corresponds to Section 2.1. Notebook “Trajectories for GSE90047” provides analysis of single cell data used in the paper. There are also several other notebooks devoted to MST and ElPiGraph trajectory inference algorithms as well as their benchmarking.

## Figures and Tables

**Figure 1 entropy-22-01274-f001:**
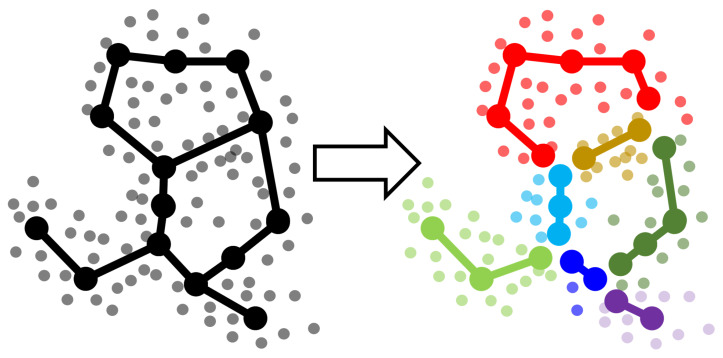
Decomposing a graph into non-branching segments and partitioning the data accordingly to the graph segments. Segments are defined as paths with ends at branching or leaf nodes. Branching nodes belong to several segments—so for better illustration here it is emphasized by multiplexing branching nodes and slightly diverging these copies from their origin (it is for illustration only). Toy example of a graph approximating a cloud of data points (shown in grey), and using its decomposition into non-branching segments for partitioning the data which defines clustering of the data cloud.

**Figure 2 entropy-22-01274-f002:**
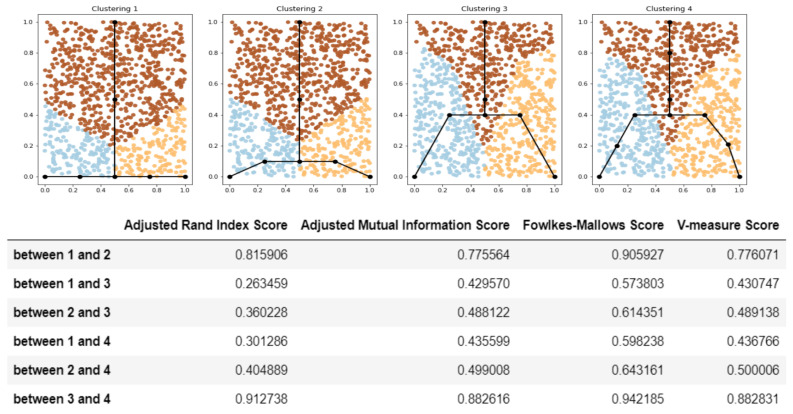
Example—trajectory comparison results. Trajectories which are close to each other from visual point of view are close by introduced similarity scores (as well as induced clusterings are visually similar). For example: trajectory 1 is visually similar to 2, but less to 2 and even less to 3,4, the scores in the table shows the same ordering. Despite graphs 3 and 4 are quite different in the sense of graph theory (three nodes are added to graph 3 to obtain graph 4), but trajectories for them are almost identical, and thus clusterings are similar and their similarity scores are quite high. That illustrates the main desired property—scores depend on trajectories not graphs themselves.

**Figure 3 entropy-22-01274-f003:**
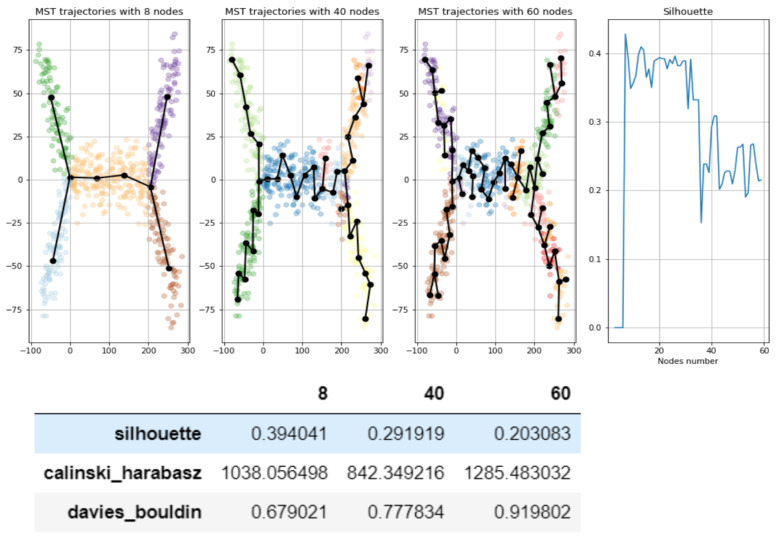
Example of estimating trajectories with the help of unsupervised clustering scores—better trajectories—get higher scores. Consider three graphs approximating the dataset of dumbell like shape with 2 branching points. We choose graphs such that the first graph fits nicely the structure of the dataset, while graph with 40 nodes fits dataset in worse way (it has several branching nodes which does not correspond to underlying dataset) and the last graph is even worse (i.e., has even more branching nodes). Thus the good score should assign best score to the first, second to the second, etc. Silhouette and Davies-Bouldin scores fit that expectation: the first graph (8 nodes) has the best scores, the second (40 nodes) the second best and the third (60 nodes) is the worst. While Calinski–Harabasz index assigns the best score to the third graph, which is counter-intuitive, nevertheless it correctly orders the first and the second graphs. (The choice of concrete graphs (and nodes numbers 8,40,60) is somewhat arbitrary the only purpose is to illustrate that scores fit the intuition—good score—corresponds to good approximating graph). Thus the figure indicates that unsupervised scores can be used to select the best graph; however, that should be done with certain care. The rightest plot confirms the same conclusion in the other way. We plot silhouette scores values for all values of nodes between 6 and 60 is given (graphs are constructed by Minimal Spanning Tree (MST) technique). It is clear that large node number correspond to lower scores on that plot, and that fits the expectations; graphs with large number of nodes constructed by MST will create many branch points which does not fit the given dataset, which has only 2 branching points. Thus it again confirms the principle—bad scores—bad approximating property.

**Figure 4 entropy-22-01274-f004:**
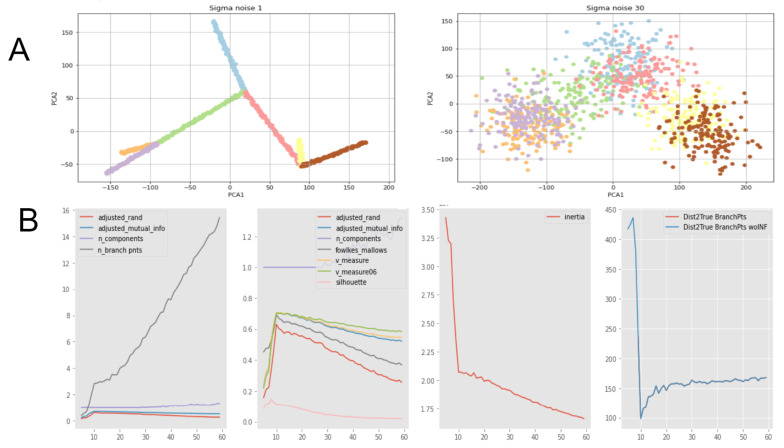
(**A**) Examples of datasets with known underlying graph structure, a symmetric binary tree with 7 edges. Datasets are 20-dimensional, visualized with PCA. The first one is with low level of noise and the second one with high level of noise. Note that despite that the noisy dataset may visually seem quite difficult for the correct graph approximation, it is distortion of 2-dimensional visualization; in original 20-dimensional space correct graph approximation is not that unlikely as one can see here. (**B**) Demonstration of agreement of our scores and naive scores like number and mutual distance to branching points, etc. On these plots X axis is number of nodes for approximate graph construction. The main observation is that ten nodes is the best value from all points of view: from our scores (the second subplot-adjusted Rand index, adjusted mutual information, Fowlkes–Mallows, etc.) as well as from the comparing distances between ground truth branching points and approximate branching points (the forth subplot), as well as elbow rule for inertia (the third subplot), as well as number from looking on number of branching points (the first subplot); ground truth has three branching points; approximate construction has three branching points for when number of nodes equal to 10 (first subplot gray line). All plots are average values over 100 randomly generated datasets having the underlying graph ground truth structure of the same shape, symmetric binary trees with 7 edges (3 branch points). See provided notebooks for details.

**Figure 5 entropy-22-01274-f005:**
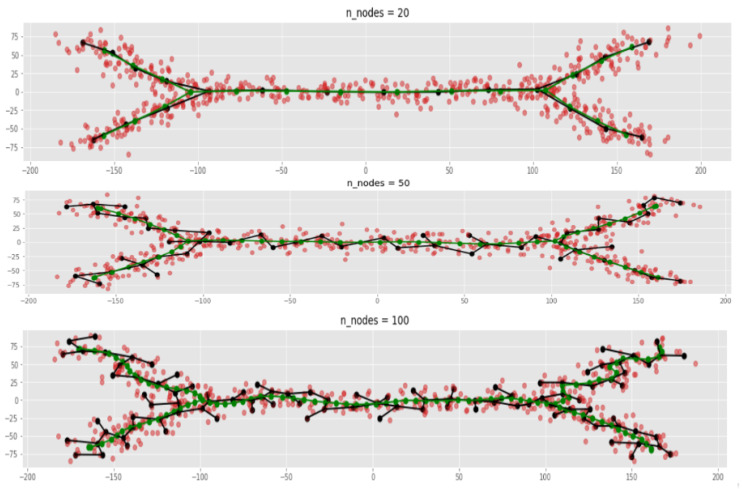
Comparison of ElPiGraph (green) and MST (black) trajectories. ElPiGraph produces correct trajectories for much wider range of main input parameter (nodes numbers) then MST.

**Figure 6 entropy-22-01274-f006:**
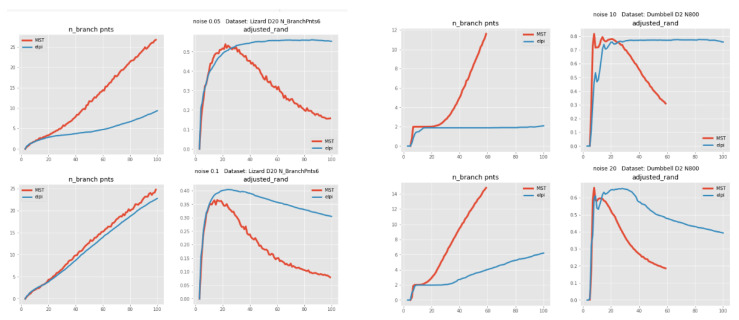
Comparison of scores for MST (red) and ElPiGraph (blue), depending on number of nodes in graph (X-axis). Simulations on 4 different types of datasets with known ground truth, results are averaged over 100 random simulations of each type (i.e., each point on each plot is result of averaging over 100 simulations). For each dataset two types of plots are presented, number of branch points and adjusted Rand index comparing trajectories of ElPiGraph and MST with ground truth. Best Rand scores for both methods are approximately the same (except left bottom case where ElPiGraph gives higher score), but the advantage of ElPiGraph is much higher stability, wide range of parameters with near highest score.

**Figure 7 entropy-22-01274-f007:**
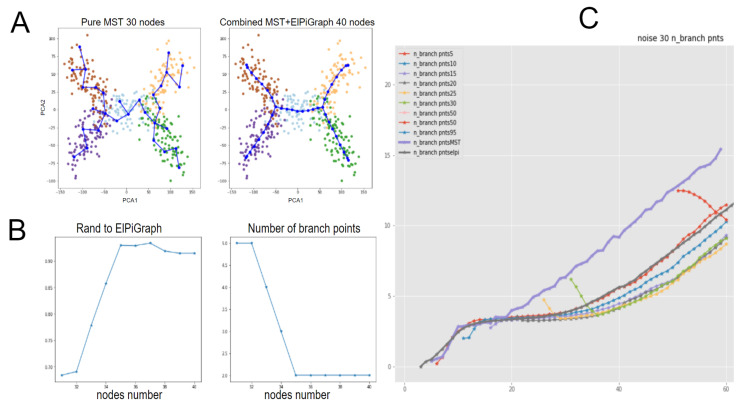
ElPiGraph forgets the initialization. **A.** Two constructions of graphs. Right: combined construction (MST+ElPiGraph) produce result similar to pure ElPiGraph despite it started from quite a different (overbranching and non-smooth) MST graph (left). **B.** Analysis how fast ElPiGraph forgets the initialization. Dynamics of scores for each step of ElPiGraph processing which started from MST graph with 30 nodes and finishing on graph with 40 nodes. X axis, number of nodes. One can see that in 5 steps scores stabilize. ElPiGraph forgets the initialization in 5 steps (for that dataset). **C.** Demonstration how fast ElPiGraph forgets the initialization. Results for MST, ElPiGraph and several combinations of MST+ElPiGraph differing by node number of MST intializing ElPiGraph. X-axis, number of nodes in constructed graph. Y-axis is number of branching points in constructed graph. Plots with different colors, different algorithms: gray—pure ElPiGraph (without MST initialization); violet—pure MST; yellow—combined: MST-25 nodes followed by ElPiGraph; green—combined: MST-30-nodes followed by ElPiGraph, etc. As one can see especially for green, yellow, red plots, they start from nearby violet plot (MST) quickly comes to around gray plot (pure ElPiGraph), thus demonstrating that ElPiGraph initialized by MST quickly forgets the initialization and shows results similar to pure ElPiGraph (sometimes even slightly better since for example several lines goes under gray line). Each point of the plot is obtained by averaging results of 100 simulations on random datasets of binary tree like shape with 3 branching points and 7 edges. Results for other simulations are similar.

**Figure 8 entropy-22-01274-f008:**
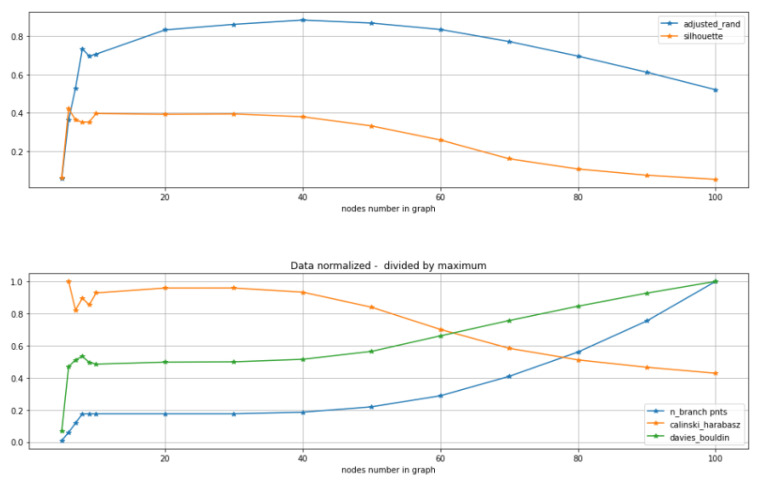
Agreement of supervised and unsupervised scores (silhouette, Calinski–Harabasz, Davies-Bouldin). For datasets with a clear ground truth graph structure (binary tree with 3 branching points) MST trajectory inference algorithm is applied. Constructed tree is compared with ground truth with adjusted Rand index based score and also scored by unsupervised score (not requiring the ground truth). (By the methodologies described above). One can see that all methods suggest that choice of parameter in the 20-40 nodes is reasonable choice. So supervised and unsupervised scores agree at this example. Thus unsupervised scores may provide an indication how to choose parameters for GBDA algorithms. Each point of the plot obtained by averaging scores on hundred simulated datasets. See notebook “SimulationUnsupervisedScoresForTI” for further details.

**Figure 9 entropy-22-01274-f009:**
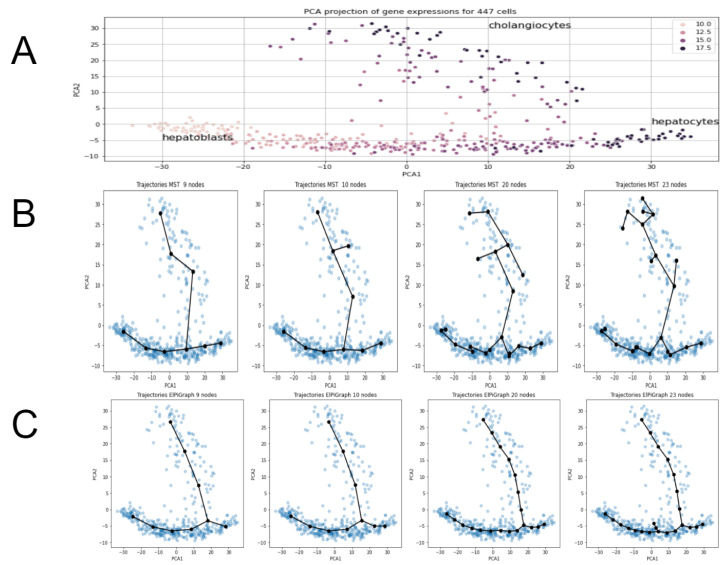
(**A**) PCA projections for single cell RNA data. Differentiation of bi-potential hepatoblasts into hepatocytes and cholangiocytes for mouse embryos. Color corresponds to different timestamps. (**B**) Trajectory inference from a single cell transcriptomic dataset using MST approach. The method produces biologically correct trajectories for a much narrower range of parameters than ElPiGraph (see next figure). Already for 10 nodes biologically incorrect branching appear. This threshold can be seen from the drop of unsupervised metrics like silhouette metric (see figures below). (**C**) Trajectory inference from a single cell transcriptomic dataset using ElPiGraph approach. The method produces biologically correct trajectories for much wider range of parameters than MST. Only for 23 nodes biologically incorrect branching appears. That threshold can be seen from drop of unsupervised metrics like silhouette metric (see next figures).

**Figure 10 entropy-22-01274-f010:**
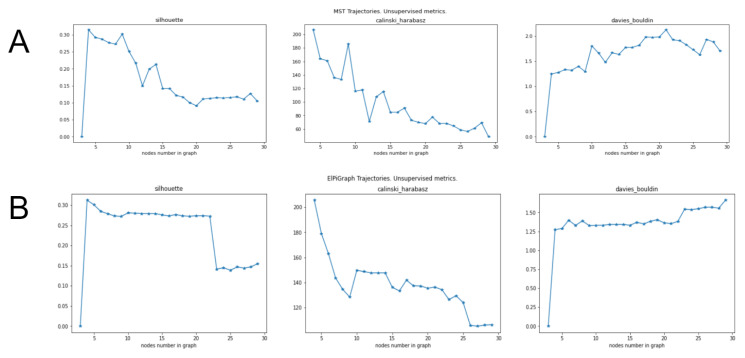
Unsupervised metrics for trajectories inferred by MST and ElPiGraph for a single cell dataset. X-axis, node number of the trajectory. (**A**) Application of MST-based approach. The metrics have extremum for 9 nodes, that point is indeed a good choice for the node number parameter. For the number of nodes more than 10, the metrics clearly indicate worse clustering quality, which corresponds to the appearance of biologically incorrect branching for 10 nodes (see previous figures). (**B**) ElPiGraph. The metrics are quite stable in wide range of node values. The clear drop of silhouette at 23 nodes corresponds to appearance of biologically incorrect branching (see previous figures). Thus it confirms that metrics can be used as an indication for the choice of parameters.

**Figure 11 entropy-22-01274-f011:**
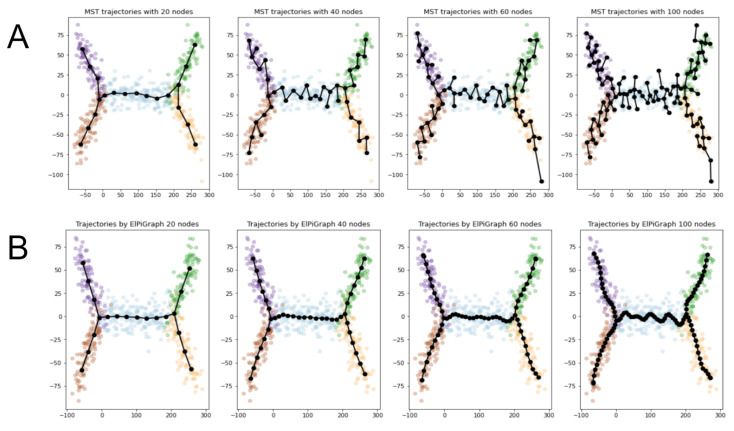
Examples of MST and ElPiGraph trajectories inference with different number of nodes. Overestimating node number would cause overbranching, creating branches which does not correspond to the ground truth. On the other hand underestimating would not allow to catch complex enough branching structure. Thus the correct choice of the number of nodes is important. (**A**) Application of MST-based approach. (**B**) ElPiGraph produces correct trajectories for much wider range of main input parameter (node number) than MST.

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
