# Peer review of "Minimum Spanning vs. Principal Trees for Structured Approximations of Multi-Dimensional Datasets"

_entropy, 2020, doi:10.3390/e22111274_

Round 1

Reviewer 1 Report

In the manuscript, the authors discuss the problem of comparison and quantification of graph-based data approximation. For this purpose, they provide a method based on graph-segmentation and comparison of the clustering quality. In general, the approach is novel and can be useful in different applied studies. My general recommendation to the authors is to reduce the number of figures and better summarize the results. Probably, some of the figures can be summarized in tables. Sometimes I found it difficult to follow the text. For example, Section 2.5 has no reference to results, and it is not clear how the reader can interpret phrases like "One can see." Some technical questions should be addressed. 1) The measure of a distance between a set and a point used by the authors may not be optimal (l. 161). It is too algorithmic, ad-hoc. Since one node at a branching belongs to several segments, there will be a sphere within which all points will have the same primary distance to different segments, and the algorithm should take another step. The authors may take another measure of distances (more standard) between a set and a point. 2) Any lemma requires formal proof. Thus, the authors can either drop it as a lemma or provide proof. 3) The example provided in Figure 2 is not clear. Since it uses random points, the result is trivial. It only calculates the distance between points in the graphs (only three change their positions). Some more complex example is required. I also would like to see here graphs with a different number of nodes. 4) For Figure 3, it would be interesting to see a figure showing the graph quality against the number of nodes from 6 to 60. I expect to see the same values for n = 6 and n = 8. Besides, it is not clear how the graphs for n = 40, 60 have been obtained. It looks like they are suboptimal, i.e., one can distribute nodes better (improving the scores). All these questions are important for understanding the method. 5) The use of multidimensional spaces for graph embedding (example in Fig. 4) also creates questions. Is there any dependence on the space dimension? 6) Some proofreading is required, and the artwork can be improved.

Author Response

We the thank the reviewer for his work and edited the text taking into account all the remarks. 

0) My general recommendation to the authors is to reduce the number of figures and better summarize the results. Probably, some of the figures can be summarized in tables.

---Reply:--- Done. Some figures are combined to subpanels, thus reducing the figure number. Results better summarized according to reviewers remarks. 

1) The measure of a distance between a set and a point used by the authors may not be optimal (l. 161). It is too algorithmic, ad-hoc. Since one node at a branching belongs to several segments, there will be a sphere within which all points will have the same primary distance to different segments, and the algorithm should take another step. The authors may take another measure of distances (more standard) between a set and a point.

---Reply:--- Done. We added several explanations to corresponding section of the text. First, indeed there can be in theory some subset of points which give rise to ambiguity , however for datasets of primarily concern - single cell data - such situations would be extremely rare and we can make arbitrary choice which would not affect the general conclusions. Second, indeed the measure we use might seem a little ad-hoc but it is done to balance the computational costs and theoretical beauty. Theoretically we can calculate proximity in slighly different way - just consider a distance from a point to a graph as closed subset of R^n thus closest point on the graph might be on the edge, not on the node (as in current implementation). However such modification would not affect much the results at least for  single cell data which are of primarily concern. We added corresponding explanations to the text. 

 2) Any lemma requires formal proof. Thus, the authors can either drop it as a lemma or provide proof.

---Reply:--- Done. We drop it. Let us nevertheless remark that the arguments presented seems to us sufficient to the reader to be convinced.  

3) The example provided in Figure 2 is not clear. Since it uses random points, the result is trivial. It only calculates the distance between points in the graphs (only three change their positions). Some more complex example is required. I also would like to see here graphs with a different number of nodes.

---Reply:--- Done. We provide more complex examples. And we added much more explanations. What is our main goal for the figure 2 - is to illustrate that our scores correspond to visually expected results, despite the random set of points is simple dataset calculation of silhouette (and other) scores for clusterings is not the thing which seems to us so much trivial. So we hope that added examples and explanations would be convincing.  

4) For Figure 3, it would be interesting to see a figure showing the graph quality against the number of nodes from 6 to 60. I expect to see the same values for n = 6 and n = 8. Besides, it is not clear how the graphs for n = 40, 60 have been obtained. It looks like they are suboptimal, i.e., one can distribute nodes better (improving the scores). All these questions are important for understanding the method.

---Reply:--- Done.  We provide a plot of silhouette for all values from 6 to 60 and explain conclusions from it. We add comments why choice 8, 40 , 60 is done - it is somewhat arbitrary - for illustration of the principle "better trajectory - better score" we need some three examples:  a) good trajectory 2) not bad 3) quite bad -  one can choose 6, 37 , 78 or whatever where the following things would be evident - look at the number of branching points - correct number is 2 - the bigger the difference - the worse - that is is intuitive measure, the score should fit that intuition - and that indeed happens as one can see from the provided results. 

5) The use of multidimensional spaces for graph embedding (example in Fig. 4) also creates questions. Is there any dependence on the space dimension?

---Reply:--- Done. Explanations are added in the discussion section. Indeed some datasets in higher dimension may not be appropriate for trajectory methods - that should be decided by the experts typically from the domain knowledge before consideration of trajectory inference algorithms. We assume that datasets we are considering to have some trajectory structure which we need to determine. 

6) Some proofreading is required, and the artwork can be improved.

---Reply:--- Done. We made some corrections. 

Reviewer 2 Report

The manuscript proposes a novel apporach to compare/benchmark different GBDA methods for data approximation using exisitng and well-established measures such as the goodness of clustering. According to my knowledge, the method is new and interesting. To illustrate the method the Authors compare performance of MST with principal graphs accross various metric, including the proposed one. In my opinon, these constitute timely and interesting results which should be published. 

Some minor technical points which the Authors may want to address in the revision: 

1) Formal statements such as "Lemma. Any graph can be uniquely partitioned into such type of segments" could be tightened. Please provide a more specific description / explanation of what the phrase "such type of segments" refers to.

2) It would be good if both, the MST data approximation algorithm and the ElPiGraph algorithm, are formally defined somewhere in the text. This will enable the reader to better understand why one method is more sensitice than the other (perhaps the Authors could point to exact steps/place in the ElPiGraph algorithm contributing to relative robustness of the ElPiGraph)

Overall, a solid and impactful contribution.

Author Response

We thank the reviewer for his remarks and edited the text taking them into account. We hope it would be convincing.

1) Formal statements such as "Lemma. Any graph can be uniquely partitioned into such type of segments" could be tightened. Please provide a more specific description / explanation of what the phrase "such type of segments" refers to.

--Reply-- we edited the text accordingly -  We decided to avoid mathematical phrasing like "lemma" since our text is of more practical nature. So we presented the arguments avoiding using "Lemmas". But still we think that idea of argument is described quite enough to fill the details, and implementation provides also kind of argument that concept works well. 

2) It would be good if both, the MST data approximation algorithm and the ElPiGraph algorithm, are formally defined somewhere in the text. This will enable the reader to better understand why one method is more sensitice than the other (perhaps the Authors could point to exact steps/place in the ElPiGraph algorithm contributing to relative robustness of the ElPiGraph)

--Reply--- We made some corrections to the text to better describe the methods. Let us mention that section "Material and Methods" contains quite a detailed description of ElPiGraph and MST, modula some facts which we refer to corresponding literature. 

Reviewer 3 Report

This paper presents an approach to compare graph-based data approximators that can also be used to fine-tune the hyper-parameters of these approximators. There are three main steps of this approach wherein the first step, the approximating graph is divided into segments, the second step is clustering the data points using these segments, and in the third step, two clustering results are compared. The authors have performed several analyses to show the efficiency of their approach. The manuscript is in the scope of this Journal and will be of great interest for 

I suggest acceptance of this manuscript.

Author Response

We thank the reviewer for his remarks.